# Improving Visual Comfort and Health through the Design of a Local Shading Device

**DOI:** 10.3390/ijerph19074406

**Published:** 2022-04-06

**Authors:** Jiao Xue, Yige Fan, Zhanxun Dong, Xiao Hu, Jiatong Yue

**Affiliations:** School of Design, Shanghai Jiao Tong University, Shanghai 200240, China; xuejiao@sjtu.edu.cn (J.X.); fan0402@sjtu.edu.cn (Y.F.); huxiao1009@sjtu.edu.cn (X.H.); jackson1997@sjtu.edu.cn (J.Y.)

**Keywords:** visual environment, glare, local shading device simulation-based design, product design

## Abstract

As people develop higher standards for the beauty of urban architecture, more and more architectural designs are exploring sources of natural lighting, such as glass curtain walls and glass domes. However, the pursuit of architectural design aesthetics introduces glare into buildings, which can be uncomfortable and even dangerous for health. Scholars in related fields have carried out many theoretical studies and design activities based on this problem. In this study, we focus on issues of glare in public buildings, aiming to improve light comfort by solving this problem. First, we propose an improved design strategy based on IDEO design thinking by adding the step of diverging from the design scheme. Second, guided by the results of a questionnaire survey and light environment simulations, we avoided the subjective simplicity of the traditional design process. Third, taking the main library of Shanghai Jiao Tong University as an example, we designed a movable sunshade that could effectively reduce glare effects and improve visual comfort, which improved the light comfort in public buildings. The simulation results show that the proposed design can be useful in buildings to effectively control glare and improve visual comfort and health.

## 1. Introduction

The research on the thermal and visual comfort of the built environment not only exists in academic research but also should be considered in modern architectural design [1,2]. Visual comfort has an impact on people’s mental state and psychological feelings [3]. For many places of production, work and study, a comfortable light environment can lift people’s spirits and improve work efficiency [4,5,6]. In addition, in indoor places for rest and entertainment, a suitable light environment can create a comfortable, elegant, lively or calm atmosphere and promote the health of human circadian rhythms [7,8].

However, with the continuous improvement of the aesthetic level of urban architectural design, more and more architectural designs are incorporating glass curtain walls, glass domes and other structures in the buildings in order to ensure the aesthetic appearance of the building and the natural lighting inside. Nevertheless, the contradiction between the pursuit of architectural design and maintaining a comfortable light environment leads to a series of visual comfort problems in buildings.

Due to the lack of effective sun-shading measures, direct sunlight brings a lot of solar radiation and causes an uncomfortable glare while illuminating the room, which affects people’s indoor activities. Glare refers to the conditions that cause visual discomfort and reduce the visibility of objects due to inappropriate brightness distribution or extreme brightness contrast in space or time in the field of view [9]. This visual comfort problem caused by glare is especially obvious in public buildings. Glare can be distracting and even dangerous: it can cause annoyance, lead to eye fatigue, reduce the sharpness of vision and can even be strong enough to block vision.

We have carried out related research on this issue and examined the relevant work of experts and scholars in different disciplines. Research on the visual comfort problem is mainly divided into three categories. First, scholars in the field of visual comfort analyze the strategy of natural lighting through traditional calculation methods [10,11,12]. Some of them put forward many comprehensive evaluation indexes for glare, illumination and shadow through simulation calculation, such as the illuminance armament deviation index (IUDI), daylight glare probability (DGP), etc. [13,14]. Second, from the perspective of architectural design, some scholars believe that the glare problem can be solved by adding a sunshade structure to the exterior of the building, such as a louver system [15,16]. Third, from the perspective of industrial design, the effect of improving the comfort of the light environment can be achieved by designing shading devices independent of buildings [17].

Such studies provide a reliable theoretical basis and method of ideas for solving glare problems and improving the comfort of the light environment, which also provides a direction for our future research. In our work, we try to find innovative solutions that combine lighting simulation methods from architecture theory and design product thinking. At the same time, we output a product that can improve visual comfort by solving the glare problem in buildings.

Libraries play an irreplaceable role in learning, work and life for university teachers, students and community members. Modern libraries usually choose fully transparent floor-to-ceiling windows to meet the lighting needs and the aesthetics of the building’s facade. However, at the same time, too much lighting reflects the sunshine directly onto desktops, causing injury to the eyes due to the uncomfortable glare and affecting people’s study and work efficiency. Take the Shenzhen Library as an example, visitors often bring and open their own umbrellas in the library to block the sunlight. Such behavior is not only inconvenient, but it also affects the overall feel and atmosphere of the library environment.

In this work, we use libraries as the study subjects and select the main library of Shanghai Jiao Tong University as the site of the final simulation test and data collection. We propose a design strategy, conduct user research and perform lighting simulations to extract the actual factors affecting light comfort and finally design a new practical sunshade product. After our test simulation, the glare evaluation index was significantly reduced at multiple points throughout the day, and the visual comfort of the public environment was effectively improved.

## 2. Methods

### 2.1. Design Thinking Based on IDEO

The concept of design thinking was created by Tim Brown, president and CEO of IDEO (a design consultancy firm) [18]. It has five steps (Figure 1):
Empathize. Consider the user’s point of view to observe and feel user experience problems.Define. Define the user’s actual problem and determine the target problem.Ideate. Think of as many solutions to the problem as possible, and then find the best one.Prototype. Present the final solution selection.Test. Test the solution to see if it solves the target problem.

Design thinking is a people-oriented problem-solving method. Following its inherent design process, both professional designers and practitioners of other industries can focus on the divergence and convergence of thinking around the problem. In this thinking path, people can produce many solutions to problems and, finally, focus on a specific scheme for deliberation, practice and testing, then finally achieve the purpose of solving the target problem.

As a method to solve problems, design thinking is not limited to the field of product design, but it is an innovative strategy used by people. Design disciplines are no longer just concerned with the appearance of products—other industries are beginning to employ design thinking to help create innovative products that focus on the actual needs and experiences of users [19].

Similarly, a single approach to solving a problem in many domains is obviously very demanding. The design thinking mechanism formulated by IDEO is representative of many innovative design thinking procedures. In the actual process of problem solving, people should construct a design innovation method that is suitable for different areas of the problem domain [20].

### 2.2. User Research

#### 2.2.1. Questionnaire Analysis

Questionnaire analysis is a method that can conduct comprehensive and systematic research on a target population concerning specific questions. It is widely used in studying commercial behavior and in academic research. Through the collection of questionnaires, a large number of sample data can be obtained. SPSS and other tools are used for data collation, statistics and analysis. This form of study targets users’ attitudes towards the target problem and even their potential demands.

#### 2.2.2. Affinity Graph

An affinity graph is a common induction tool in design disciplines [21]. Affinity refers to the mutual relationship between information. By using an affinity graph, the key information of the problem can be sorted out from the complex and fragmented information so that the target problem is clear.

In our study, the process of using an affinity graph can be divided into four steps:Define the problem.Identify the problem we are trying to solve with affinity diagrams.Brainstorm.Compile all the collected information, such as questionnaire analysis results, interview results, etc., into information cards.Categorize associations.Classify these information cards into categories based on their relationships.Discuss the evaluations.Discuss and evaluate the categorized information to obtain guidance for solving the target problem.

### 2.3. Building Lighting Environment Simulation

#### 2.3.1. Site Selection for the Simulation Study

Considering architectural design and user sample size, we chose the main library of Shanghai Jiao Tong University as the location to carry out light environment simulation research. Shanghai Jiao Tong University, as an institution with a capacity of more than 30,000 students, has abundant architectural space and user research samples. Its campus public space is rich, including Bao Yugang Library, Li Zhengdao Library, the main library, a variety of teaching buildings, office buildings and other buildings. Among them, the main library we choose has a total of four floors, and each floor is set up with four areas: A, B, C and D. Due to architectural design problems, the building is always affected by the light environment, but due to the geographical location of the campus, most of the teachers and students still choose to study and work here. The promenade reading area in area D is equipped with floor-to-ceiling windows, which are always full due to the spacious and comfortable learning environment. However, this area is illuminated by the sun from early in the morning, with dazzle occurring most frequently between 10 am and 4 PM. Most users in this area adopt the strategy of carrying their own umbrellas to resist glare, but they still need to leave their seats for a short time when the glare is most intense. Due to the quiet environment of the other three areas, many users choose this area for self-study. However, those areas still generate glare from 2 pm. Taking the main library of Shanghai Jiao Tong University as the research object, it is convenient for us to obtain rich user research samples and conduct a simulation of the typical areas.

#### 2.3.2. Glare Evaluation Index—Daylight Glare Probability (DGP)

Daylight Glare Probability in building technology simulations is an indicator to describe the probability of glare occurring, hereafter referred to as DGP [9]. As a powerful glare metric, DGP can detect glare sources through the contrast of direct sunlight as well as specular reflections. Compared with other glare indicators (e.g., UGR, DGI, CGI and VCP), DGP is a newer glare indicator [22].
(1)$$\mathrm{DGP}=5.87\times 10^{-5}E_{v}+9.18\times 10^{-2}\log\left(1+\underset{i}{\sum }\begin{array}{c} L_{s,i}^{2}\omega_{s,i}\\ E_{v}^{1.87}P_{i}^{2} \end{array}\right)+0.16$$

In Formula (1):$E_{v}$ is the vertical eye illuminance (lux);$L_{s}$ is the luminance of the source (cd/m^2^);$\omega_{s}$ is the solid angle of the source;*P* is the position index.

The lower the DGP, the lower the probability of glare problems [23,24]. According to the scale of DGP (Table 1), when the DGP is below 0.35, it is at an imperceptible level.

#### 2.3.3. Validation of Simulation Tools

In this study, we conducted relevant tests of the product throughout the design process, based on the Ladybug and Honeybee platform, a collection of free computer applications supporting environmental design and education in the parameterized design plug-in Grasshopper from Rhino. We first designed the following steps to validate our simulation tools:Obtain the measuring point layout of the actual research site.We used the TES-1330A illuminance measuring instrument, TESTO hot wire anemometer, black bulb thermometer and other thermal measuring instruments to test the physical environment of the research site (the main library of Shanghai Jiao Tong University) and conducted spot measurements in the most representative reading area.Obtain the illuminance simulation layout of the research site.
(1)We first obtained the data needed for building in a simulated environment through outdoor measurements, and then we used the Rhino software to build the architectural model [25] (Figure 2).(2)Based on the Ladybug and Honeybee platform, we carried out the programming of the illuminance simulation in the software (Figure 3):
①We set the material relationship corresponding to the construction of the research carrier;②We imported the annual weather file of Shanghai; ③We simulated the building’s annual light; ④We calculated the annual light during working hours.(3)Finally, we output the illuminance simulation results of the building.Compare.

By comparing the layout between the illuminance simulation and the actual measurement point, we can obtain the validation results of the simulation tools.

#### 2.3.4. Simulation of DGP

In our study, we used DGP to describe the degree of glare problem resolution in order to evaluate the effectiveness of our developed product. Through the two plug-ins, Ladybug and Honeybee, we carried out the design and actual simulation process (Figure 4).

Assign the material for different architectural components and define their parameters.Install a virtual library user in the building.Input the weather data of Shanghai.
(1)Set the weather type as sunny.(2)Set the simulation date and time.Run the simulation and obtain the target DGP data.

## 3. Experiment and Results

In our research, in order to make the final design output more in line with the building lighting simulation results and effectively solve the glare at the theoretical level, we improved the original design process based on design thinking (Figure 5). In the original process, Step 3, ideate, obtains different kinds of solutions for target problems through brainstorming and other methods, and this process is realized by the designer’s own ability. At the same time, designers need to rely on their own perceptual judgment to select the optimal scheme to promote. In our research, whether the glare problem can be solved can actually be quantitatively evaluated by measuring and calculating the glare-related indicators, so we can select the appropriate scheme for design and promotion according to a certain amount of data. We divided the ideate step into two parts: user research and solution deduction. The user research part is based on the traditional design discipline method to carry out the user questionnaire survey and analysis and affinity graph induction. The solution deduction uses the light environment simulation tool to simulate and propose the target scheme.

### 3.1. User Research

#### 3.1.1. Questionnaire Research

We designed the questionnaire for this study based on the fact that a large number of teachers and students at the main library of Shanghai Jiao Tong University work in direct sunlight while holding umbrellas. We gathered 239 valid questionnaires.

We used the questionnaires to record the behavior patterns and subjective feelings of teachers and students in the library. The interviewees were classified according to whether or not they were holding umbrellas in the library and whether their locations were exposed to direct sunlight or indirect sunlight. The questionnaire is divided into two parts. The first part collects the basic information of the interviewees, including their attire, and the second part collects their seating preferences and reasons for seat selection (Figure 6).

An analysis of the data showed that 29.8% of the subjects believed that direct sunlight does not affect them (Table 2 and Table 3). Moreover, most students chose to study in the area affected by direct sunlight, mainly for the favorable natural lighting conditions and beautiful scenery. Furthermore, under the condition of direct sunlight, more than half chose to change seats, showing the importance of improving the seating area affected by glare.

#### 3.1.2. Design Requirements and Function Sorting

From the analysis of the questionnaire, we obtained unclassified information fragments (Figure 7), which expressed subjective intentions from the perspective of users. We used design tools to sort out the requirements and functions of these information fragments.

We chose an affinity diagram as the combing tool that could put the insights and observations of the designer on paper and carry out targeted information aggregation [21]. The affinity diagram (Figure 8) is based on the specific observation results and information of this study [17], dividing the acquired information fragments into three parts: user experience, actual conditions and user needs.

### 3.2. Design Evolution

#### 3.2.1. Verification Results of Simulation Tools

From the questionnaire, we obtained preliminary design requirements for shading equipment. Our research simulated the library environment, and the final shading effect of the product also needed to be tested by simulation. In order to ensure the credibility of the final test results, we measured the illuminance of the actual library building and compared it with the simulation results of the software.

Illumination (i.e., light intensity) refers to the energy of visible light received per unit area, which is used to indicate the intensity of light and the amount of illumination of the surface area of an object.

In step 1 of the experiment, based on the user research, the measurement area included two areas of the library: A400 and the 4F atrium (Figure 9). We selected the two areas and conducted the actual point measurement.

Our indoor measurement focused on the distribution of illuminance in space at a certain moment in the room. Figure 10 and Figure 11 show our illuminance measuring point layout diagram.

With the work of the simulation platform, we output the illuminance simulation result of the building and visualize it as Figure 12.

By comparing the illuminance simulation with the data of the corresponding point of the actual measurement point, we obtained the comparison results of the actual measurement and the simulation as well as the fitting equation:y = 1.1107x − 661.08(2)

R^2^ = 0.84 (R^2^ = 0.84 > 0.75)

This indicates that the simulation software we used had sufficient accuracy, which ensures the credibility of our product testing.

#### 3.2.2. Design Evolution

According to the DGP scale (Table 1), when the DGP is below 0.35, we can deduce that the glare problem has been solved well and the light environment has been significantly improved [9]. Therefore, the present work defines DGP decline as the research goal and uses this as a basis to judge the effectiveness of the designed product. Based on the library model established in the illuminance simulation link, we selected the A400 reading room as the simulation object.

First, we configured the materials and defined their parameters for the different components of the constructed building model in the program. We used the default values of the program to set the reflection coefficients of the walls, ceilings, columns, floors and desktops of the study areas and the light transmittance of the windows.

After the material definition of the simulated environment was complete, we installed a virtual library user in the building, simulating and setting the perspective of his learning and working behavior to ensure that he sat at a window, setting his line of sight to 1.1 m from the floor [26]. Then, we input the weather data of Shanghai. Since our research focus was the glare problem, we set the weather type as sunny, and we set 30 December 2020 as the date for the simulation. From 13 to 17 o’clock on this day, we ran simulations every hour and gathered our target DGP data.

We present the data here with visual diagrams. Table 4 shows the DGP data of the research carrier from 15 to 18 o’clock on 20 December without the shading device. With no shade, the DGP index remained above 0.7 until sunset, which was extremely uncomfortable for human eyes.

The next idea of our design evolution was to add sunshades to the architectural simulation to block the sun rays, and we used the program to simulate and calculate the DGP of different schemes and compare them with the results without sunshades [27].

Based on the indoor light environment, we set the light environment without sunshade as group A. Based on the rectangular sunshade with a width of 0.6 m, we designed the strategy group B0. From the height of the sunshade and the included angle between the sunshade and the user, we carried out data iteration. Finally, we proposed five optimization design strategies, from B1 to B5 (Table 5) and compared four groups of DGP data to find suitable product design elements (Table 6).

As shown in Figure 13, compared with control group 1, for the basic strategy of group B0, the sunshade essentially did not have an impact on the glare problem, as the DGP did not differ significantly from group A.

In control group 2 (Figure 14), we did not change the included angle between the sunshade and the user but gradually increased the height of the sunshade. As the height increased to 0.9 m, the DGP decreased significantly from 1.00 to 0.32 at 13:00 o’clock. The DGP was less than 0.35, which was comfortable enough for the user. At the same time, we found that when the height was increased again, the influence of the sunshade on DGP was no longer significant.

In control group 3 (Figure 15), when the height of the sunshade was 0.9 m and the angle between the sunshade and the user was rotated to 30°, the DGP decreased significantly from 1.00 to 0.31 at 14:00, achieving the target DGP value.

Based on the above three control groups, we obtained rough constraints in the subsequent product design and implementation process. In particular, having part of our sunshade equipment below 0.6 m was ineffective. Only when the height of the equipment was appropriate could the sunshade task be performed. At the same time, the angle between the appropriate sunshade equipment and the user also had a positive effect on reducing DGP.

### 3.3. Product Definition

The core goal of our research was to prevent direct sunlight from hitting people’s faces and desktops and to reduce the discomfort of the glare caused by direct sunlight [26,28]. Based on the above analysis, we mapped the requirements for the functionality of the product and designed it [29]. In order to achieve an effective shading effect, we framed the height of the product to 100 cm and limited its range of motion to 40–100 cm in the vertical direction. Furthermore, and from a functional point of view, the product must have a mechanical structure that can move freely to match the changing angle of the sun. In all, the product needs to meet three requirements:
Beautiful and integrated into the environment: it does not impact the overall artistic quality of the built environment and does not affect the perception of the participants in the building.
Practical and easy to operate: without additional learning costs to users, it can be operated simply and intuitively.Intelligent and convenient storage: when not in use, it does not occupy space and can be telescopic for storage.

## 4. Product Design and Implementation

### 4.1. Introduction to the Product

After analyzing the results, we developed a portable 180° mobile shading device. We designed a mobile sunshade device that is 40 cm long and 5 cm in diameter when folded up and 1 m high when fully opened. The product has three states: a fully closed state, a half open state and a fully open state (Figure 16).

#### 4.1.1. Product Main Structures

The product is composed of four parts, from bottom to top: a fixed base, control panel, telescopic main rod and shading fan (Figure 17).

The fixed base mainly has the role of fixing the position of the shade, and its upper part is connected with the telescopic main rod (Figure 18). When using the desktop card in its depression, by adjusting the knob on the base, one can control the upper and lower part of the base spacing, so as to clamp the desktop and have the product fixed on the desktop (Figure 19).

As the interactive interface between the user and the product, the control panel is how the user actually operates the product (Figure 20). It is attached to the surface of the telescopic main rod. The control panel has two functions: it can control the expansion and contraction of the product, and it can display information, such as the use time and weather.

The lower part of the telescopic main rod is connected with the fixed base, and the upper part is connected with the sunshade fan through the movable component (Figure 21). The telescopic main rod has two levels. The length of the contraction state is 40 cm, and, when extended, the secondary rod extends upward, with a total length of 70 cm. The lower part of the rod has a control motor. The rod is controlled by the control panel and can be adjusted to extend the height in practical use within the range of 40–70 cm.

The shading fan is connected by a movable component and a telescopic main rod (Figure 22). The shading fan has two forms of contraction and expansion. When retracted, the shading fan drops down and is close to the main rod. When unfolded, a sunshade is formed (Figure 23). The movable component can rotate around the telescopic main rod, thus rotating the sunshade fan. At the same time, the active component is equipped with a control motor, which is controlled by the control panel. In practice, it is controlled to rotate in the range of [−90°, 90°] through the control panel.

#### 4.1.2. Interactive Display of Product

The product is operated through the control panel, which is divided into three areas (Figure 24): the upper part is the display area, the middle is the control area and the lower part is the switch button.

In practice, the user first fixes the product in the appropriate table position through the fixed base. Then, the user long presses the power button on the control panel to wake up the product.

Through the control area of the control panel, the user can press the up and down keys to control the expansion main rod part of the product to expand or retract the sunshade. If they touch the switch button again for a short time, the shading fan will fully open. Users can push the left and right keys to change the angle rotation of the sunshade.

The upper display area of the control panel shows the product’s time of use and the weather conditions of the day.

#### 4.1.3. Control Wireframes of Product

The entire product control is commanded through the screen. The product has two control motors, located in the lower part of the main rod and the upper part of the main rod in the movable components. Two motors enable the expansion and rotation of the product. The control wireframes of the product are shown in Figure 25.

#### 4.1.4. Product Superiority

From the perspective of user experience, compared with traditional sunshade devices, the actual sunshade area of our product is 70 cm above the desktop. This leaves sufficient space to ensure the shading effect, preserve the view of the user and maintain natural light and views of scenery outside the window. Our design reduces the invasiveness of sun-shading products for users and thus improves the user experience [30]. Based on the product functions, we placed the interactive interface at the bottom of the product to ensure that users can easily control and adjust the product while they are studying or working. The screen also displays weather information. Starting from the actual shading effect, as our product design is based on the simulation results of light environment, the product can effectively reduce the DGP and improve users’ light comfort while not affecting the appearance of the building or the demand for indoor lighting.

### 4.2. Product Simulation

After finishing the product design, we carried out the convergence step of the design process and simulated the actual shading effect of the design. In the simulation software, we built a rod with an adjustable height between 0.4 and 0.7 m and a semicircle with a radius of 0.3 m that can rotate 180° with the rod as the axis on the upper part of the rod. We used this simulation to test the efficiency of our design.

By adjusting the height and rotation angle of this component, we simulated the DGP value from 13:00 to 17:00 o’clock on 20 February and 30 December [17]. According to the results, the shading effect of our product was further improved.

As can be seen in Figure 26 and Figure 27, we conducted two comparisons on the two target dates to compare the DGP values with and without the shading device. On 20 February (Figure 26), when our product was set up in the simulation environment, the measured DGP was controlled below 0.41 between 13:00 and 15:00 and had a downward trend between 16:00 and 17:00, with a value of 0.21–0.76.

On 30 December (Figure 27), when our product was set up in the simulation environment, the measured DGP was controlled below 0.36 between 13:00 and 15:00 and had a downward trend between 16:00 and 17:00, with a value of 0.21–0.68.

## 5. Discussion

In future work, we will be committed to providing a more intelligent product design generation method, optimizing the target-oriented algorithm based on the architectural light environment theory and proposing the scientific path of product design. At the same time, our ultimate goal is to explore how to reduce DGP. In addition, we will further improve the product from the perspective of design, considering the product attributes of the design results so that it has more industrial aesthetics and more functions. Our products should be more intelligent and greener so as to improve users’ experience and enhance their psychological state.

## 6. Conclusions

In our research, we ultimately aim to enhance the light comfort and health of the built environment. We focus on issues of glare in public buildings, aiming to improve light comfort by solving such problems. The current work has the following two innovations.

First, we propose an improved design strategy based on IDEO design thinking, by adding the step of diverging from the design scheme. Guided by the results of questionnaire surveys and light environment simulations, we avoided the subjective simplicity of the traditional design process.

Second, we introduced a glare evaluation index, DGP, through which we carried out a series of light environment simulations and obtained reliable data to support our product design. This lead step makes our design strategy more reliable than the traditional design process. We combined the user research method of design with the light environment simulation theory so that the design of the product is no longer determined by the designer but guided by data from the very beginning, which ensures the efficient shading effect of the final product.

Third, our research resulted in a portable sunshade product that can rotate 180°. The design of the product meets our research objective: to improve light comfort by reducing the DGP to solve glare problems. In addition, the design of the product takes into account the user experience, which is one of its advantages. While solving the user’s actual needs, their workspace is unobstructed, and their experience is enhanced.

In our final simulation test, our product was able to reduce the DGP from 1 to less than 0.41 between 13:00 and 15:00 on the two target dates selected. Thus, the validity of the designed product and the rationality of the optimized design strategy are verified.

There are some limitations to our research. First, from the perspective of design strategy, although this design process is guided by simulation results, it is not closely combined with theories related to the light environment, so it cannot be completely problem-oriented. Second, from the perspective of product design, our product design still has room for improvement in the user experience. Third, according to the target results, our product has not yet achieved the optimal improvement in light comfort and has not achieved a significant decrease in DGP for the whole day.

## Figures and Tables

**Figure 1 ijerph-19-04406-f001:**
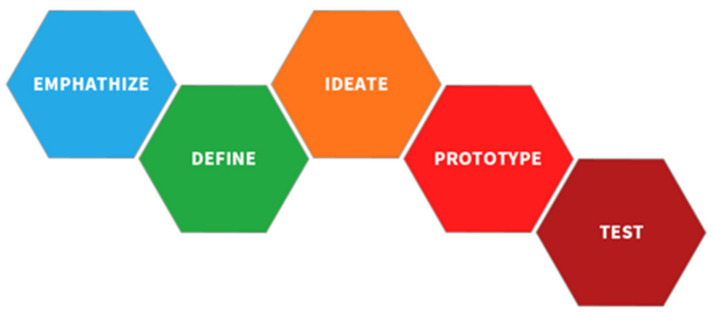
Five steps of design thinking proposed by Tim Brown.

**Figure 2 ijerph-19-04406-f002:**
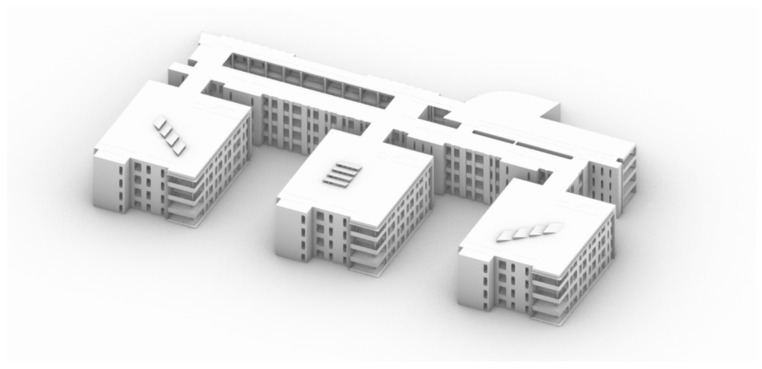
Simulation model of the main library of Shanghai Jiao Tong University.

**Figure 3 ijerph-19-04406-f003:**
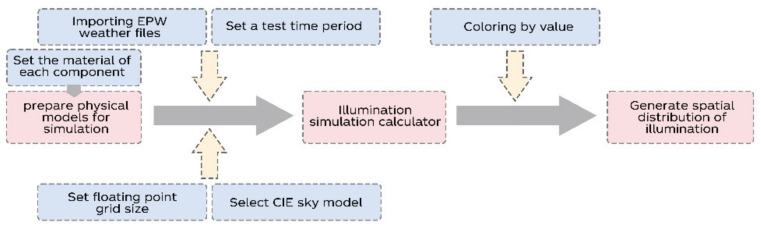
The programming of illuminance simulation.

**Figure 4 ijerph-19-04406-f004:**
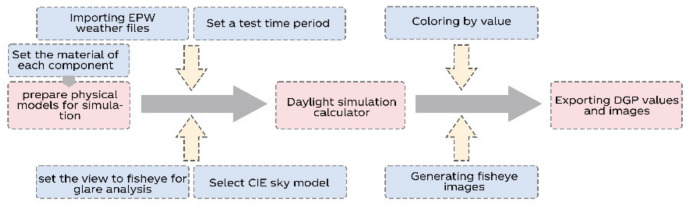
The programming of the DGP simulation.

**Figure 5 ijerph-19-04406-f005:**
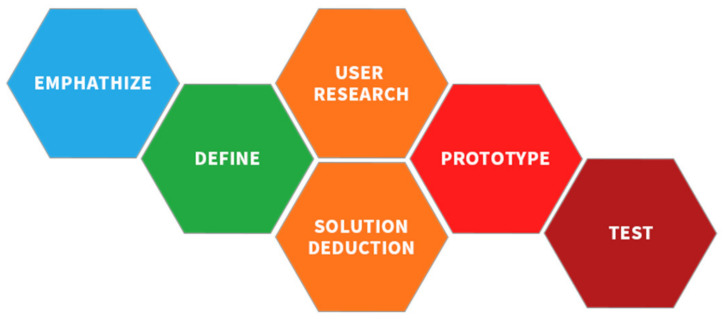
Improved design thinking.

**Figure 6 ijerph-19-04406-f006:**
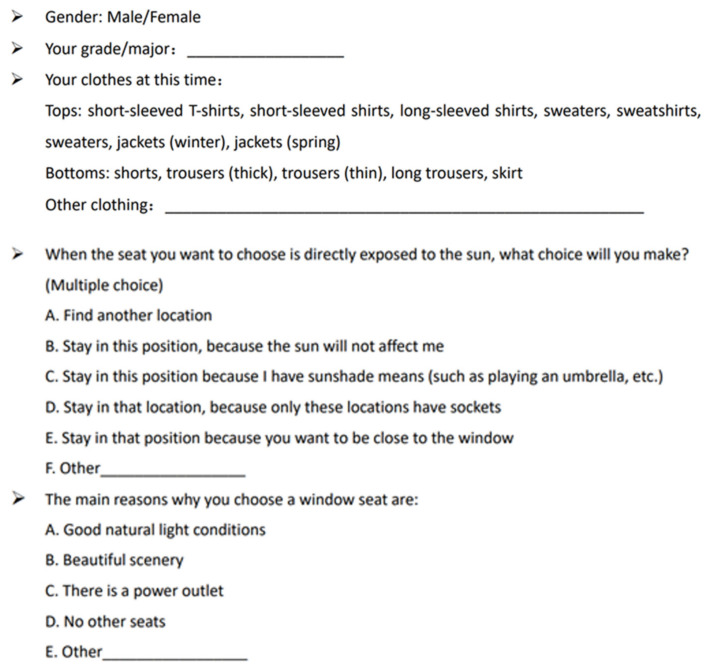
Part of questionnaire.

**Figure 7 ijerph-19-04406-f007:**
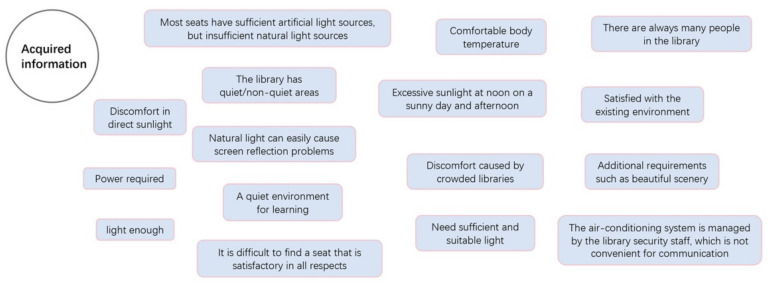
Unclassified information fragments.

**Figure 8 ijerph-19-04406-f008:**
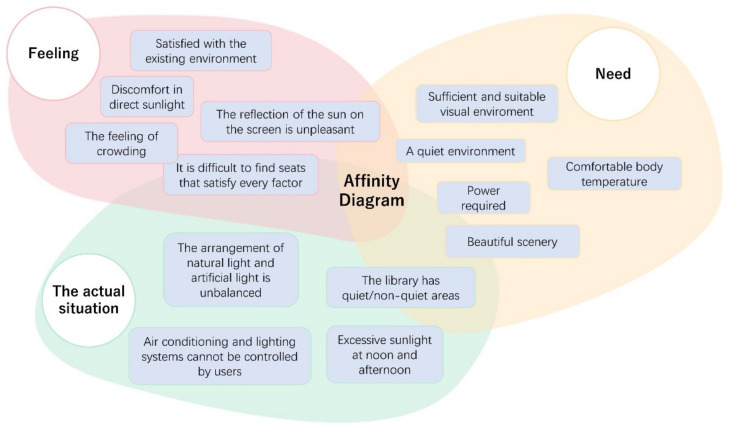
The affinity diagram.

**Figure 9 ijerph-19-04406-f009:**
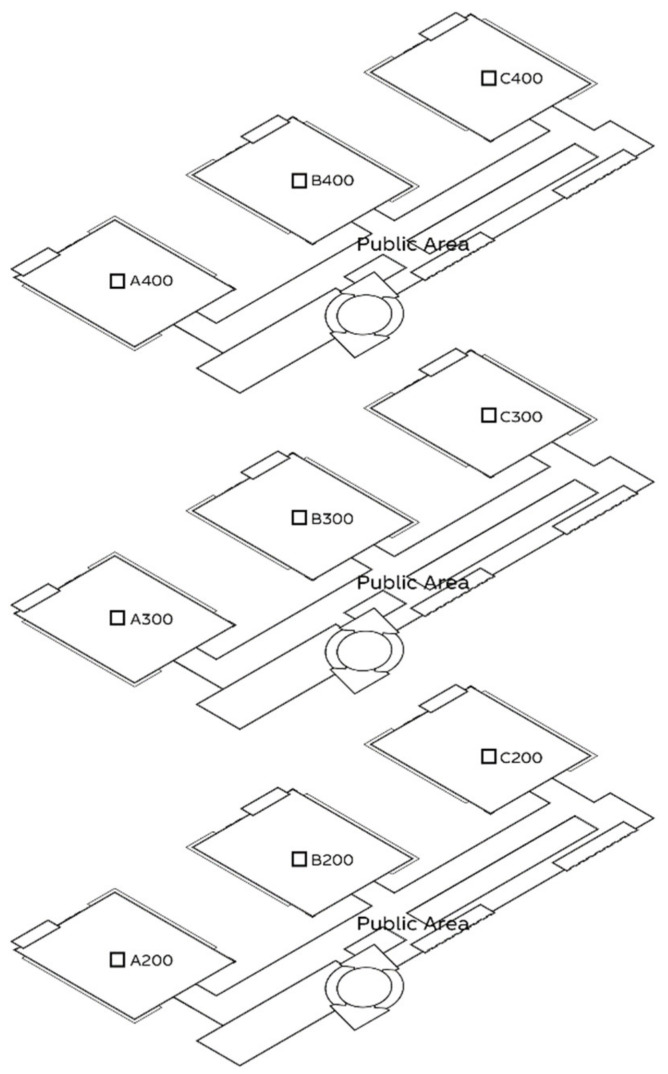
Regional distribution map of Shanghai Jiao Tong University Library.

**Figure 10 ijerph-19-04406-f010:**
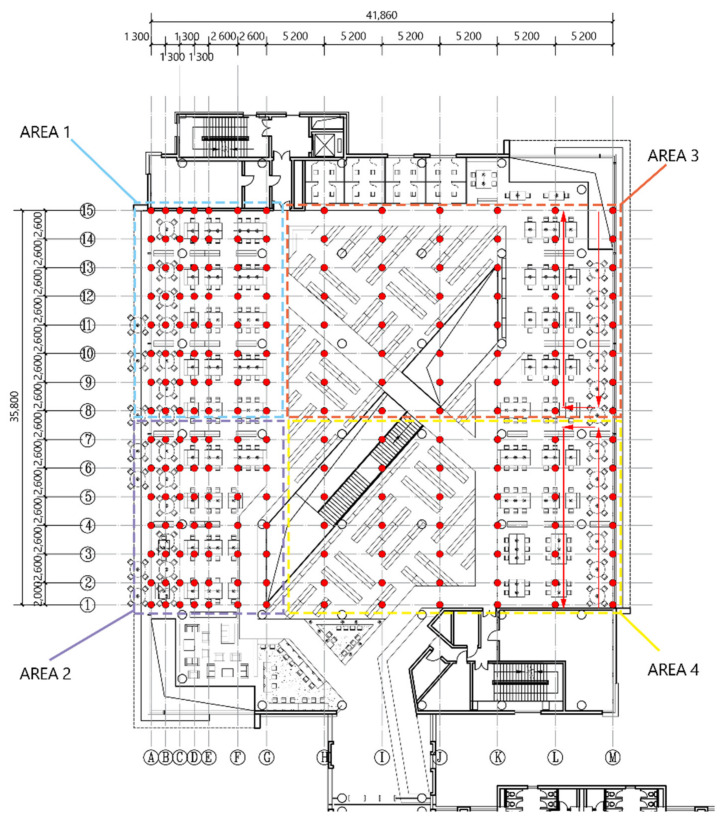
The measuring point layout of A400.

**Figure 11 ijerph-19-04406-f011:**
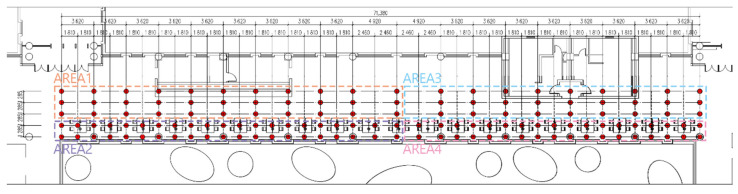
Measuring point layout in the atrium on the fourth floor.

**Figure 12 ijerph-19-04406-f012:**
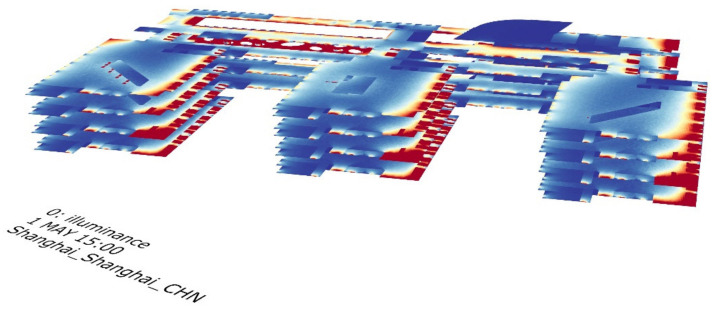
Illuminance simulation result.

**Figure 13 ijerph-19-04406-f013:**
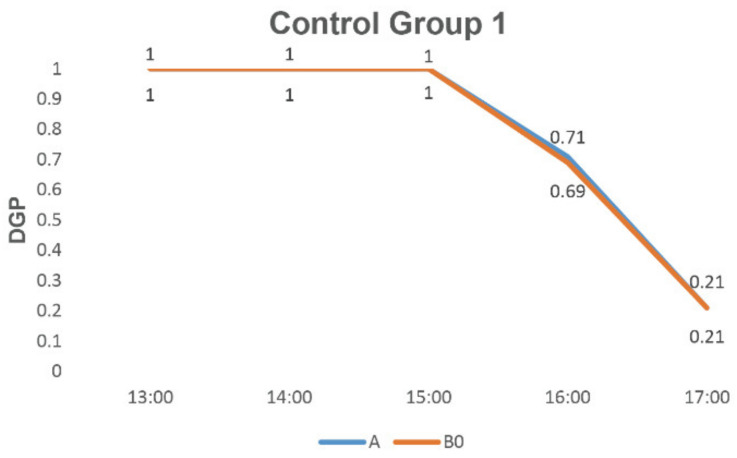
Comparation results of control group 1.

**Figure 14 ijerph-19-04406-f014:**
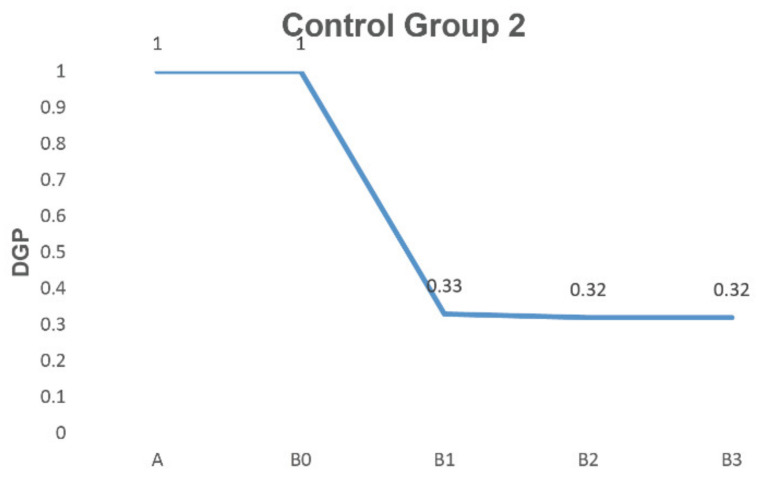
Comparation results of control group 2.

**Figure 15 ijerph-19-04406-f015:**
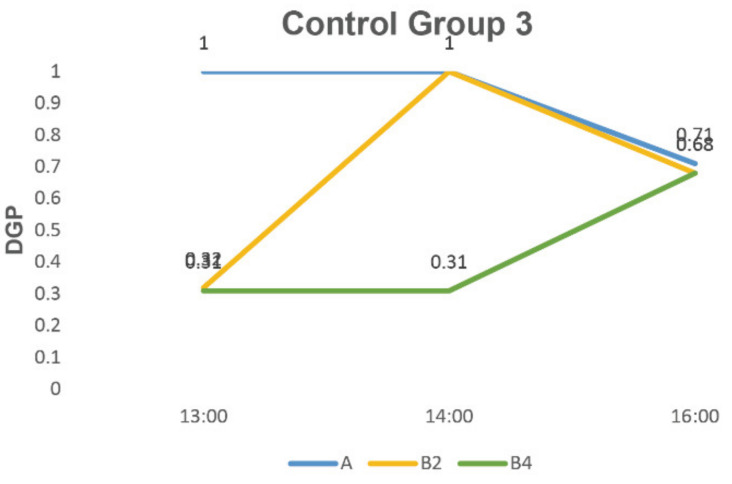
Comparation results of control group 3.

**Figure 16 ijerph-19-04406-f016:**
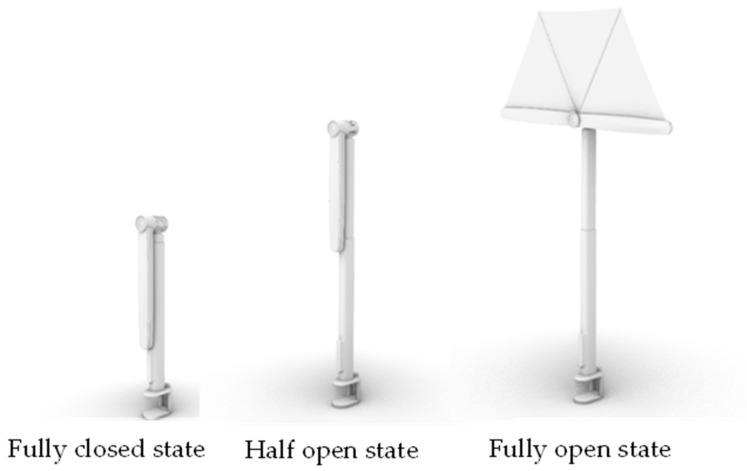
Three states of product.

**Figure 17 ijerph-19-04406-f017:**
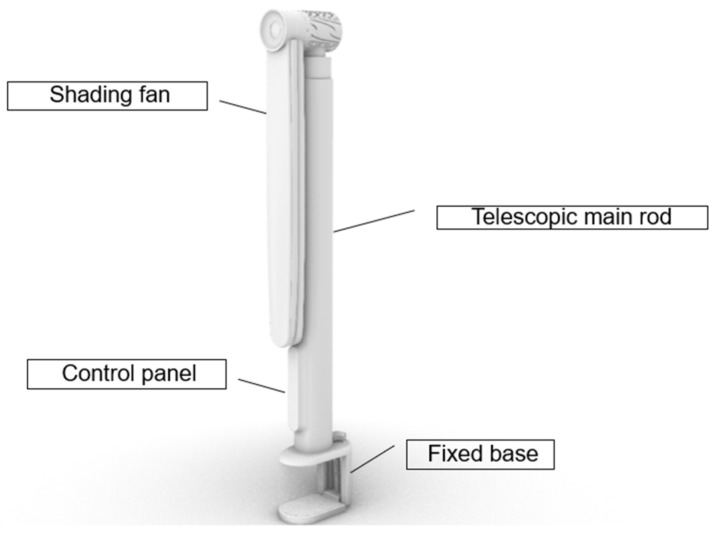
Portable 360° mobile shading device.

**Figure 18 ijerph-19-04406-f018:**
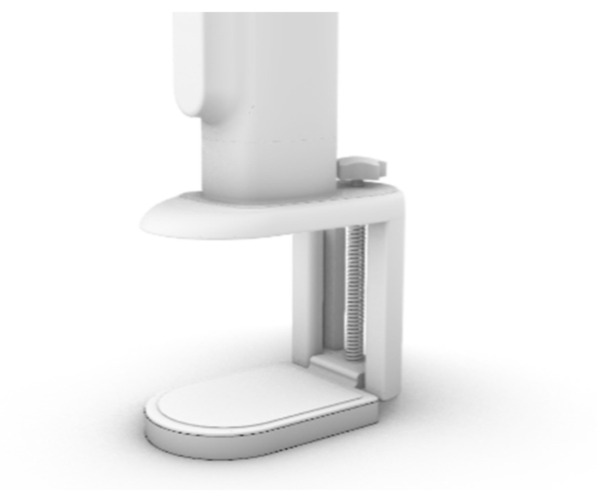
Fixed base.

**Figure 19 ijerph-19-04406-f019:**
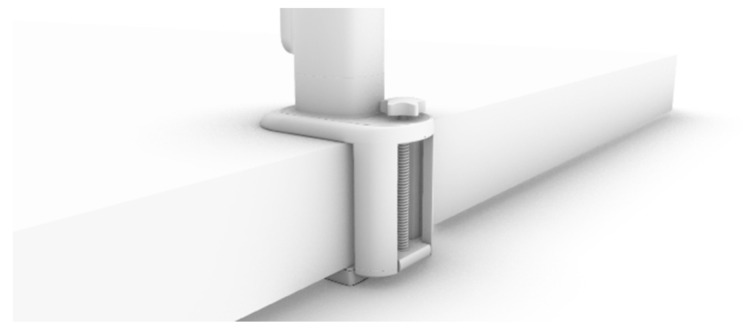
Fixed with desk.

**Figure 20 ijerph-19-04406-f020:**
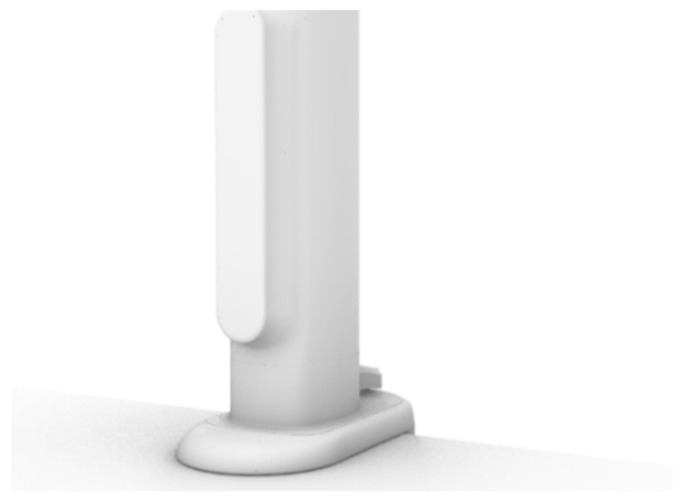
Control panel.

**Figure 21 ijerph-19-04406-f021:**
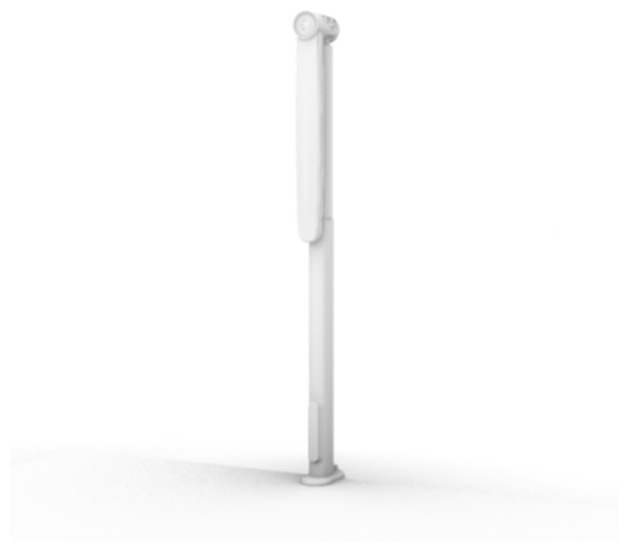
Telescopic main rod.

**Figure 22 ijerph-19-04406-f022:**
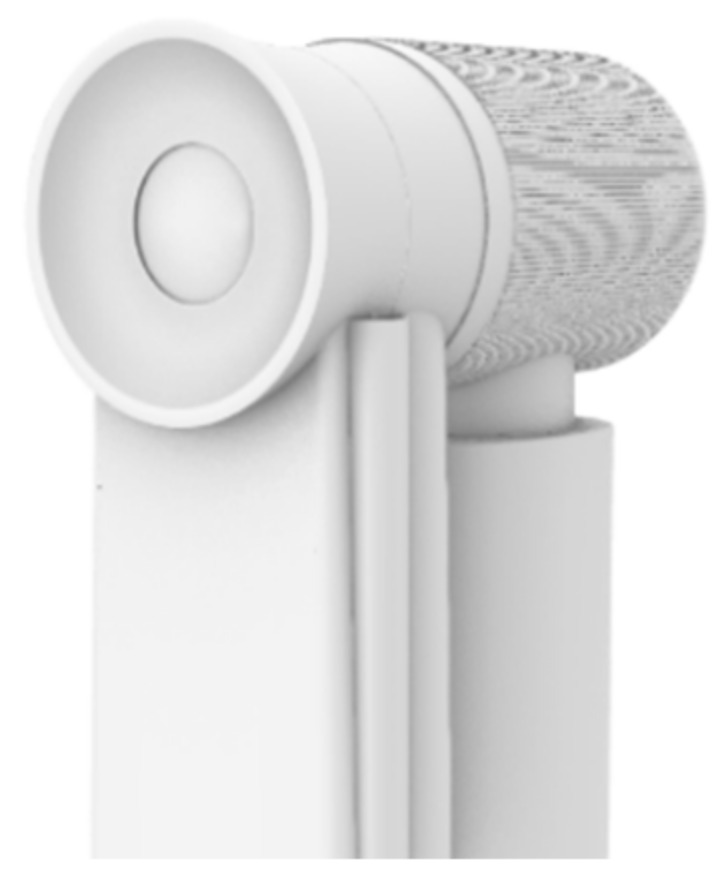
Shading fan and movable component.

**Figure 23 ijerph-19-04406-f023:**
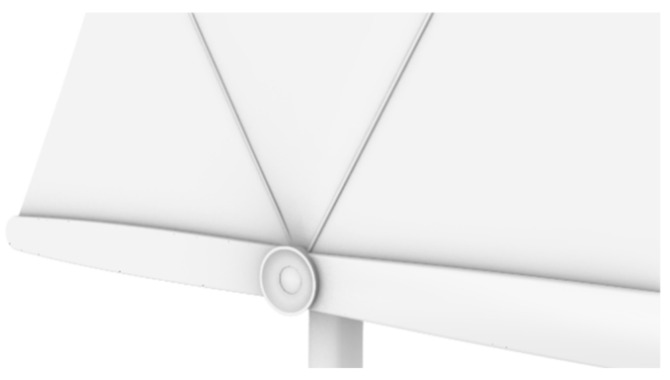
Unfolded shading fan.

**Figure 24 ijerph-19-04406-f024:**
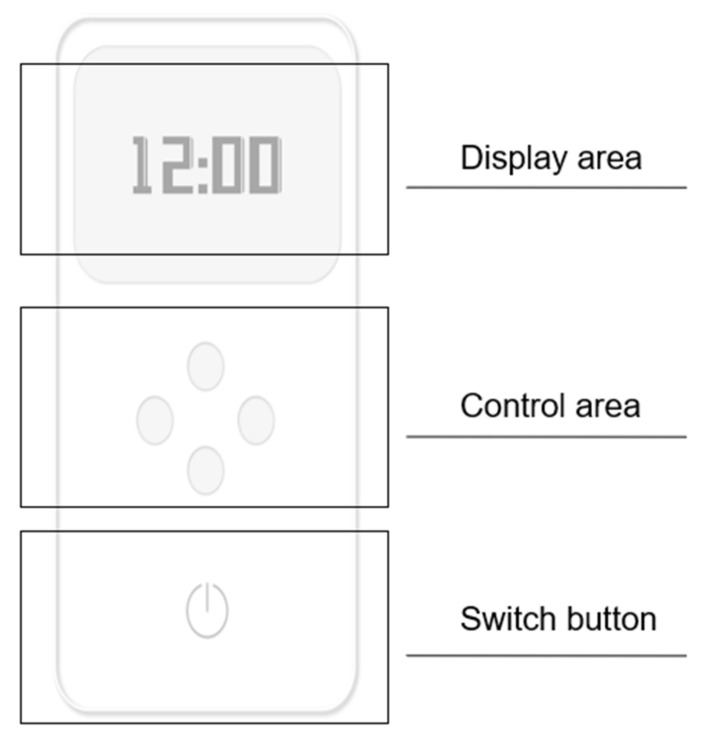
Function division of the control panel.

**Figure 25 ijerph-19-04406-f025:**
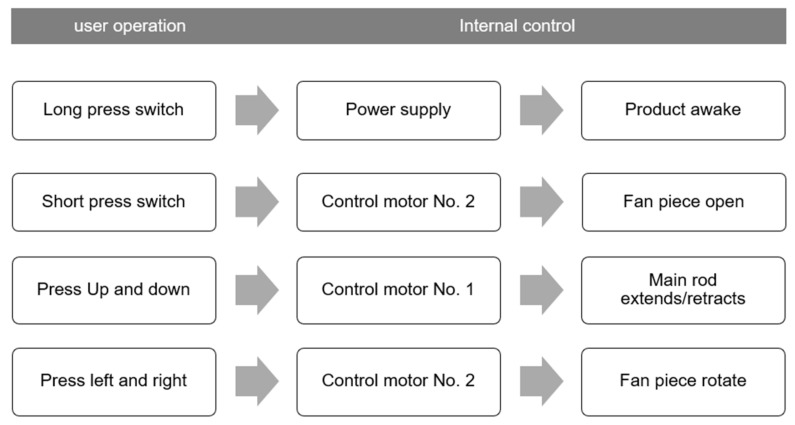
Control wireframes of product.

**Figure 26 ijerph-19-04406-f026:**
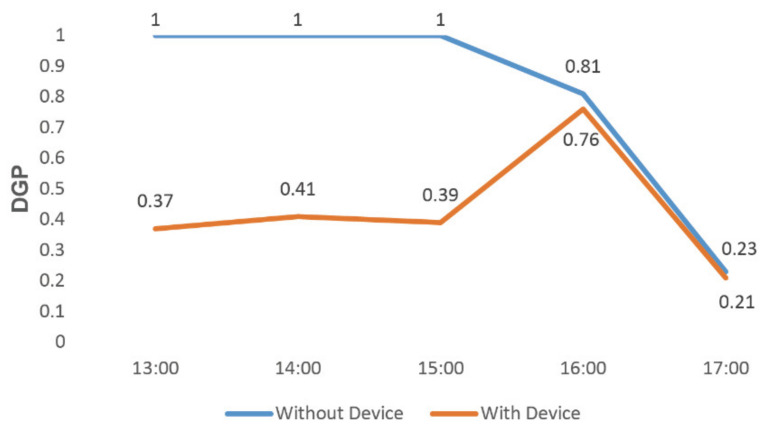
Product simulation on 20 February.

**Figure 27 ijerph-19-04406-f027:**
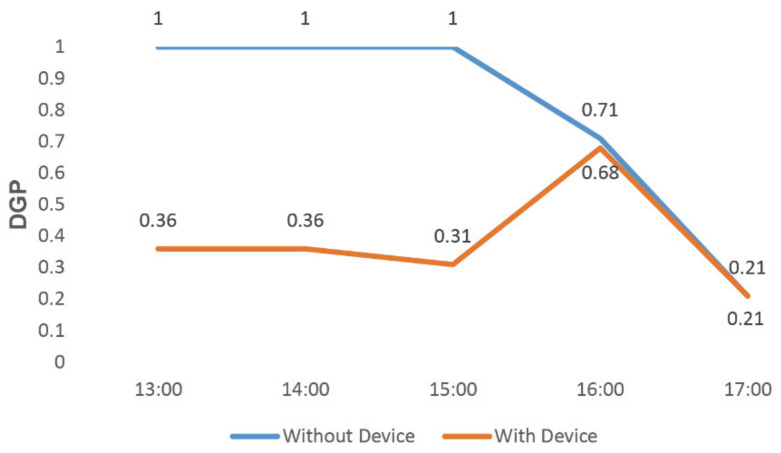
Product simulation on 30 December.

**Table 1 ijerph-19-04406-t001:** DGP classification.

DGP Level	DGP Value
Intolerable Glare	0.45 ≤ DGP
Disturbing Glare	0.40 ≤ DGP< 0.45
Perceptible Glare	0.35 ≤ DGP< 0.40
Imperceptible Glare	DGP< 0.35

**Table 2 ijerph-19-04406-t002:** Why you choose a window seat.

	The Response		Percentage of Cases
	Number of Cases	Percentage	
Good natural light conditions	87	30.00	45.50
Beautiful scenery	52	17.90	27.20
power outlets	131	45.20	68.60
No other seats	14	4.8	7.3
Others	6	2.1	3.1

**Table 3 ijerph-19-04406-t003:** When the seat you want is directly exposed to the sun, what choice will you make?

		The Response		Percentage of Cases
		Number of Cases	Percentage	
Find another location		126	38.30	52.9
Stay in this position	because the sun will not affect me	71	21.6	29.8
	have sunshade means (such as umbrella, etc.)	32	9.70	13.4
	For power outlets	68	20.70	28.6
	For near to the window	28	8.50	11.8
	Other	4	1.20	1.7

**Table 4 ijerph-19-04406-t004:** DGP on December 20 without shading device.

	13:00	14:00	15:00	16:00
DGP Without Shade	1.00	1.00	1.00	0.71

**Table 5 ijerph-19-04406-t005:** Optimization design strategies.

	Height of the Prototype (m)	Angle between the Prototype and User (°)
A	0	0
B0	0.6	0
B1	0.8	0
B2	0.9	0
B3	1.0	0
B4	0.9	30
B5	1.0	30

**Table 6 ijerph-19-04406-t006:** Comparation strategy.

	Group
Control group1	A, B0
Control group2	A, B0, B1, B2, B3
Control group3	A, B2, B4

## Data Availability

Not applicable.

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
