# Peer review of "Improving Visual Comfort and Health through the Design of a Local Shading Device"

_ijerph, 2022, doi:10.3390/ijerph19074406_

Round 1
Reviewer 1 Report
Building facades comprise the first layer between the occupant and the outside world. They are largely responsible for conserving building energy, encouraging occupant comfort, providing views, and allowing for the expression of design aesthetics. The trend toward sustainable architecture has inspired an increasing number of transparent building facades. There are various issues that can occur from excessive natural light, one of which is "intolerable glare". Glare directly affects human activities that use spaces such as these as working spaces, creation spaces, and relaxing spaces, yet this is particularly so in working spaces since people must focus their eyesight.
In the submitted manuscript, the authors (Xue et al.) try to find innovative solutions that combine lighting simulation methods from architecture theory and design product thinking. At the same time, they output a product that can improve visual comfort by solving the problem of glare in buildings. The core goal research was to prevent direct sunlight from hitting people's faces and desktops, and to reduce the discomfort of the glare caused by direct sunlight.
Comments and suggestions:
Assessing the originality of the research problem addressedit should be noted to the authors presented some of the results in 2021: Xue, J., et al. (2021). "Smart Design of Portable Indoor Shading Device for Visual Comfort—A Case Study of a College 504 Library." Applied Sciences 11(22): 10644. https://doi.org/10.3390/app112210644
- Line 142: Table 1 published [in:] Xue, J., et al. (2021). "Smart Design of Portable Indoor Shading Device for Visual Comfort—A Case Study of a College 504 Library." Applied Sciences 11(22): 10644. https://doi.org/10.3390/app112210644
- Line 178 – Figure 4 published [in:] Xue, J., et al. (2021). "Smart Design of Portable Indoor Shading Device for Visual Comfort—A Case Study of a College 504 Library." Applied Sciences 11(22): 10644. https://doi.org/10.3390/app112210644
- Line 233 – Figure 8 published [in:] Xue, J., et al. (2021). "Smart Design of Portable Indoor Shading Device for Visual Comfort—A Case Study of a College 504 Library." Applied Sciences 11(22): 10644. https://doi.org/10.3390/app112210644
- Lines 267 and 278: Error! Reference source not found’
- Figure 3, 4, 10, 11- are illegible.
- Please complete the manuscript with the latest literature
- Sun Y, Liu X, Qu W, Cao G, Zou N. Analysis of daylight glare and optimal lighting design for comfortable office lighting. 2020; 206:164291.
- Sankaewthong S., Horanont T., Miyata K., Karnjana J. Designing a facade by biomimicry science to effectively control natural light in buildings (Glare analysis) Materials Science and Engineering. 2021;1148: 012002 doi:10.1088/1757-899X/1148/1/012002
- Reference Lines 487-524: References should be described as follows (Instructions for Authors - IJERPH), depending on the type of work:
Example - Journal Articles:
Author 1, A.B.; Author 2, C.D. Title of the article. Abbreviated Journal Name Year, Volume, page range.
Author Response
Dear Editors and Reviewers:
The revised version of the manuscript (Manuscript ID: ijerph-1602613) has been submitted, which has been cautiously revised according to the reviewers’ comments.
We would like to express our sincere gratitude to the Editor, Associate Editor and three anonymous reviewers for their efficient works and constructive suggestions. Their insightful comments are very useful in improving the quality and presentation of this paper. The authors also thank the Editor and Associate Editor for giving us the opportunity to revise this manuscript. All the comments have been seriously considered and addressed in the revised manuscript.
Comments and suggestions:
Assessing the originality of the research problem addressedit should be noted to the authors presented some of the results in 2021: Xue, J., et al. (2021). "Smart Design of Portable Indoor Shading Device for Visual Comfort—A Case Study of a College 504 Library." Applied Sciences 11(22): 10644. https://doi.org/10.3390/app112210644
- Line 142: Table 1 published [in:] Xue, J., et al. (2021). "Smart Design of Portable Indoor Shading Device for Visual Comfort—A Case Study of a College 504 Library." Applied Sciences 11(22): 10644. https://doi.org/10.3390/app112210644
- Line 178 – Figure 4 published [in:] Xue, J., et al. (2021). "Smart Design of Portable Indoor Shading Device for Visual Comfort—A Case Study of a College 504 Library." Applied Sciences 11(22): 10644. https://doi.org/10.3390/app112210644
- Line 233 – Figure 8 published [in:] Xue, J., et al. (2021). "Smart Design of Portable Indoor Shading Device for Visual Comfort—A Case Study of a College 504 Library." Applied Sciences 11(22): 10644. https://doi.org/10.3390/app112210644\
Response:
We appreciate the reviewer for this constructive comment.
The articles mentioned in points 1-3 are a study conducted by the same research group. We are from the research group of solar thermal environment in public spaces of Shanghai Jiao Tong University. Under the leadership of professor Xue, we are conducting research on indoor light environments and thermal environments in public spaces and other built environments.
This article is led by professor Xue, with the participation of Yibo Wang and Mingxiang Wang. Their main work was to discover glare problems in public spaces and try to verify their functionality with simple products. Their main work was to discover glare problems in public spaces and try to verify their functionality with simple products. The product alleviated the problem at a specific moment, providing a boost for subsequent research. The innovation of our research this time is that we use design thinking combined with simulation to guide product design with new design paths and develop products that can effectively solve problems most of the time. This research is still led by professor Xue, with the participation of researchers from multidisciplinary backgrounds such as Yige Fan, Xiao Hu, Jiatong Yue, etc.
The table1 proposed in point 1 is a measurement index mainly used in light environment research. We decided to keep it after discussion, this indicator can help scholars and readers understand our research better.
Figure 4 mentioned in point 2, this picture is a schematic diagram when we use the simulation software, the original intention is to help the readers to understand. Thanks again for pointing out here, we've replaced this diagram with a more clearly thought-out flow chart.
Figure 8 mentioned in point 3 is a common tool for sorting out ideas when using design thinking to carry out design behaviors. We hope to use it to show the main problems encountered by users in public spaces. At the same time, its existence is also reflected that we have combined the design method in this research and carried out the innovation of the research path. We have modified it. To avoid ambiguity, a citation has been added at line259.
- Lines 267 and 278: Error! Reference source not found
Response:
We appreciate the reviewer for this comment.
We have carefully checked the manuscript and fixed this problem.
- Figure 3, 4, 10, 11- are illegible.
Response:
We appreciate the reviewer for this constructive comment.
Based on previous comments, we have replaced figure3 and figure4 with more understandable flow charts. And figure10 and figure11 have been replaced.
- Please complete the manuscript with the latest literature
- Sun Y, Liu X, Qu W, Cao G, Zou N. Analysis of daylight glare and optimal lighting design for comfortable office lighting. 2020; 206:164291.
- Sankaewthong S., Horanont T., Miyata K., Karnjana J. Designing a facade by biomimicry science to effectively control natural light in buildings (Glare analysis) Materials Science and Engineering. 2021;1148: 012002 doi:10.1088/1757-899X/1148/1/012002
Response:
We appreciate the reviewer for the recommendation.
The suggested literature has been added, as can be found in Line 185, Page 5 and Line 317,Page 11.
- Reference Lines 487-524: References should be described as follows (Instructions for Authors - IJERPH), depending on the type of work:
Example - Journal Articles:
Author 1, A.B.; Author 2, C.D. Title of the article. Abbreviated Journal Name Year, Volume, page range.
Response:
We appreciate the reviewer for the comment.
The reference has been reformatted according to IJERPH’s requirement.
Reviewer 2 Report
- Your manuscript needs some revision in both grammatical.
- ABSTRACT: abstract is adequate, but need to be edit and reformed.
- Key words should be corrected and should be revised.
- Introduction: should add some new references published in International Journal of Environmental Research and Public Health and other similar journal.
- Materials and Methods: please add the time duration of study.
- Materials and Methods: Please describe how the location of study was selected, in details.
- Discussion: The discussion part should modify.
- Discussion: Suggest adding a paragraph on directions for future research, practice and policy.
- Conclusion: there is differing between conclusion of abstract and full text. Also, conclusion should be edited.
Author Response
Dear Editors and Reviewers:
The revised version of the manuscript (Manuscript ID: ijerph-1602613) has been submitted, which has been cautiously revised according to the reviewers’ comments.
We would like to express our sincere gratitude to the Editor, Associate Editor and three anonymous reviewers for their efficient works and constructive suggestions. Their insightful comments are very useful in improving the quality and presentation of this paper. The authors also thank the Editor and Associate Editor for giving us the opportunity to revise this manuscript. All the comments have been seriously considered and addressed in the revised manuscript.
Your manuscript needs some revision in both grammatical.
Response:
We appreciate the reviewer for the comment.
We have used a professional service to edit the manuscript. Below please find the editing certificate:
- ABSTRACT: abstract is adequate, but need to be edit and reformed.
Response:
We appreciate the reviewer for the comment.
We have revised the abstract.
- Key words should be corrected and should be revised.
Response:
We appreciate the reviewer for the comment.
The keywords have been changed to: visual environment, glare, local shading device, simulation-based design, product design.
- Introduction: should add some new references published in International Journal of Environmental Research and Public Health and other similar journal.
Response:
We appreciate the reviewer for this comment.
The following recently published paper has been added to the introduction.
Baloch, R.M., Nichole Maesano, C., Christoffersen, J., Mandin, C., Csobod, E., de Oliveira Fernandes, E., Annesi-Maesano, I. and Sinphonie Consortium, 2021. Daylight and School Performance in European Schoolchildren. International Journal of Environmental Research and Public Health, 18(1), p.258.
Nagare, R., Woo, M., MacNaughton, P., Plitnick, B., Tinianov, B. and Figueiro, M., 2021. Access to Daylight at Home Improves Circadian Alignment, Sleep, and Mental Health in Healthy Adults: A Crossover Study. International journal of environmental research and public health, 18(19), p.9980.
Osibona, O., Solomon, B.D. and Fecht, D., 2021. Lighting in the Home and Health: A systematic Review. International journal of environmental research and public health, 18(2), p.609.
- Materials and Methods: please add the time duration of study.
Response:
We appreciate the reviewer for this comment.
This study is one of the results of the Research Laboratory of Public Space visual and thermal Environment at Shanghai Jiao Tong University. The study began last March and has been going on ever since.
- Materials and Methods: Please describe how the location of study was selected, in details.
Response:
We appreciate the reviewer for this constructive comment.
Based on the consideration of architectural design and user sample size, we chose the main library of Shanghai Jiao Tong University as the location to carry out light environment simulation research. As an institution with a capacity of more than 30,000 people, Shanghai Jiaotong University has a wealth of architectural space and user research samples. The research site we chose is affected by the light environment all the year round due to architectural design issues, but due to the special geographical location, most teachers and students still choose to study and work here. We have added section 2.3.1 to the manuscript to fully illustrate this point.
- Discussion: The discussion part should modify.
Response:
We appreciate the reviewer for the suggestion.
We have revised the discussion part.
- Discussion: Suggest adding a paragraph on directions for future research, practice and policy.
Response:
We appreciate the reviewer for the suggestion.
We have moved the future study part in the Conclusion to the Discussion section. In the future, we will be committed to providing a more intelligent product design generation method, optimizing the target-oriented algorithm based on the architectural light environment theory and proposing the scientific path of product design.
- Conclusion: there is differing between conclusion of abstract and full text. Also, conclusion should be edited.
Response:
We appreciate the reviewer for the suggestion.
The conclusion was drawn based on the analyses of the main results, and the abstract was a summary of the entire manuscript.

Reviewer 3 Report
The authors write about a device designed by them that can improve visual comfort.
The article is ambitious and interesting. There are some points that it would be good for the authors to clarify better and also make some corrections that would help improve the quality of the article.
1. The authors state that the pursuit of aesthetics in architectural design introduces glare into buildings, which can be uncomfortable and even dangerous to health. The authors provide solutions to the issue of glare through a divergence strategy and surveys; in addition to lighting simulations. Could the authors provide quantitative values ​​on the reduction values ​​in glare levels.
2. Suggested device can be helpful in all use or activity. Why have they not contemplated results taking into account different lighting systems or types of luminaires? How can the results be extrapolated to other uses, activities or lighting systems?
3. What are the hours selected for the simulations?
4. How does the lighting environment affect the results?
5. How far the device can be developed; and what future work the authors foresee regarding the device. Justify this answer in the conclusions.
6. How can the improvement obtained in lighting comfort be quantified and to what extent can the results be improved?
7. It is possible to improve the results with algorithms; but to what extent have surveys reflected an improvement in the psychological state of people?
8. The authors have considered the issue of respecting circadian rhythms in the search for health and well-being. Justify how the circadian rhythms of people have been considered or add the implication in this theme of the article.
9. Enrich the state of the art with the following impact articles.
Wagiman, K. R., Abdullah, M. N., Hassan, M. Y., & Mohammad Radzi, N. H. (2021). A new metric for optimal visual comfort and energy efficiency of building lighting system considering daylight using multi-objective particle swarm optimization. Journal of Building Engineering, 43. https://doi.org/10.1016/j.jobe.2021.102525
Ma, G., & Pan, X. (2021). Research on a visual comfort model based on individual preference in China through machine learning algorithm. Sustainability (Switzerland), 13(14). https://doi.org/10.3390/su13147602
Rabani, M., Madessa, H. B., & Nord, N. (2021). Building retrofitting through coupling of building energy simulation-optimization tool with cfd and daylight programs. Energies, 14(8). https://doi.org/10.3390/en14082180
Moyano, D. B., Fernández, M. S. J., & Lezcano, R. A. G. (2020, May 1). Towards a sustainable indoor lighting design: Effects of artificial light on the emotional state of adolescents in the classroom. Sustainability (Switzerland). IPDM. https://doi.org/10.3390/su12104263
10. Quantitatively justify whether the device has an ecological impact or energy savings.
11. Improve the quality of figures 3 ,4
12. Increase the size of figures 8,9,10,11 and 12
13. Improve the quality of graphs 13,14,15,26 and 27; eliminating the background lines on the scale and in the frame.
Author Response
Dear Editors and Reviewers:
The revised version of the manuscript (Manuscript ID: ijerph-1602613) has been submitted, which has been cautiously revised according to the reviewers’ comments.
We would like to express our sincere gratitude to the Editor, Associate Editor and three anonymous reviewers for their efficient works and constructive suggestions. Their insightful comments are very useful in improving the quality and presentation of this paper. The authors also thank the Editor and Associate Editor for giving us the opportunity to revise this manuscript. All the comments have been seriously considered and addressed in the revised manuscript.
The authors write about a device designed by them that can improve visual comfort.
The article is ambitious and interesting. There are some points that it would be good for the authors to clarify better and also make some corrections that would help improve the quality of the article.
1. The authors state that the pursuit of aesthetics in architectural design introduces glare into buildings, which can be uncomfortable and even dangerous to health. The authors provide solutions to the issue of glare through a divergence strategy and surveys; in addition to lighting simulations. Could the authors provide quantitative values ​​on the reduction values ​​in glare levels.
Response:
We appreciate the reviewer for the question.
In the validation of the simulation tools, we used architectural measurement equipment such as illuminance measurement instruments to measure different measurement points in the study area. These quantitative data reflect light comfort and glare problems in the area without product occlusion At the same time, we have tested it on a small scale using the initial version of the product prototype, and after occlusion, the measurement results have been reduced. Thanks again for your inquiry, our next research work is to make an actual product prototype and verify the solution of the glare problem through quantitative data measurement.
Suggested device can be helpful in all use or activity. Why have they not contemplated results taking into account different lighting systems or types of luminaires? How can the results be extrapolated to other uses, activities or lighting systems?
Response:
We are very grateful for the valuable questions on equipment expansion.
First, we chose the library as a representation of the public space. Taking our research carrier, the main library of Shanghai Jiaotong University as an example, its lighting system is very stable and the use of user groups has been considered at the early stage of design. The glaring problem is the discomfort caused by light, which in this type of building is mainly caused by sunlight. In the current research, we are still in an active design, focusing on specific research carriers to carry out the design. As the development direction pointed out by line509 in the manuscript, our next research work will try to reverse thinking and deduce the design through the algorithm. At present, this study has been carried out and cannot be described in detail due to space limitations. We believe that with further research, we can obtain findings that can be broadly applicable to different architectural spaces and lighting systems.
What are the hours selected for the simulations?
Response:
We appreciate the reviewer for this constructive comment.
When measuring the light environment, different dates and times will produce different results. The simulations developed in our study serve two main purposes. The first point is to verify whether our product is effective or not. These two dates are days with relatively high sunshine intensity, which can effectively simulate the effect. The second point is to verify whether the products designed under the guidance of the new research path have better effects. In the literature cited in manuscript line447, these two dates were also chosen when the results were obtained. So we also chose these two dates for simulation, for better comparison and verification.
How does the lighting environment affect the results?
Response:
We appreciate the reviewer for the comment.
The artificial lighting environment generally does not interfere with the result, since the major glare source is the sun. The outdoor daylight environment also does not have influence. The device was designed for use in the sunny days. When the daylight outside the building was cloudy or overcast sky, the shading device is not needed.
How far the device can be developed; and what future work the authors foresee regarding the device. Justify this answer in the conclusions.
Response:
We appreciate the reviewer for the comment.
We have added the future study in the discussion: In future work, we will be committed to providing a more intelligent product de-sign generation method, optimizing the target-oriented algorithm based on the architectural light environment theory and proposing the scientific path of product design. At the same time, our ultimate goal is to explore how to reduce DGP. In addition, we will further improve the product from the perspective of design, considering the product attributes of the design results, so that it has more industrial aesthetics and more functions. Our products should be more intelligent and greener, so as to improve users' experience and enhance their psychological state.
How can the improvement obtained in lighting comfort be quantified and to what extent can the results be improved?
Response:
We appreciate the reviewer for this constructive comment.
As mentioned in 2.3.2 of the manuscript, in our research ideas, we introduced a measure called DGP. Daylight Glare Probability in building technology simulations is an indicator to describe the probability of glare occurring, hereafter referred to as DGP.
In the existing studies, the DGP evaluation indexes that are generally accepted by scholars have been pointed out in table1. Therefore, one of the main ideas for judging whether light comfort is improved in our study is that light comfort is improved when glare, the factor causing the decline of light comfort, is solved. The metric that quantifies this process is DGP. Our simulation quantifies the result by figuring out if the DGP goes down.
The degree of improvement is reflected by DGP value. In the absence of research intervention, the value of DGP reaches 1, which, in the measurement index proposed in table1, already belongs to the glare that human eyes cannot tolerate. However, after the intervention of the study, the DGP value dropped below 0.45, which was a comfortable level.
It is possible to improve the results with algorithms; but to what extent have surveys reflected an improvement in the psychological state of people?
Response:
We are grateful to the reviewers for their valuable questions.
Because our research combines design thinking, we use a lot of design methods in the process of development. During the field interview, we contacted many students who were troubled by glare. More than half of them had to use umbrellas in the library to solve the problem of light environment. This question was also confirmed by the data in the questionnaire. At present, when using the product prototype to carry out the test, the feedback of the experiencer is positive, believing that the experience is improved. As stated in the conclusion,in line473,we are currently developing the user feedback of large volume, and will optimize the product design based on this in the next research.
The authors have considered the issue of respecting circadian rhythms in the search for health and well-being. Justify how the circadian rhythms of people have been considered or add the implication in this theme of the article.
Response:
We are grateful to the reviewers for the suggestion.
When we improve the comfort of the light environment, we can also improve the regularity of circadian rhythm, thus improving physical and mental health. We've introduced a related article at Line30.
Enrich the state of the art with the following impact articles.
Wagiman, K. R., Abdullah, M. N., Hassan, M. Y., & Mohammad Radzi, N. H. (2021). A new metric for optimal visual comfort and energy efficiency of building lighting system considering daylight using multi-objective particle swarm optimization. Journal of Building Engineering, 43. https://doi.org/10.1016/j.jobe.2021.102525
Ma, G., & Pan, X. (2021). Research on a visual comfort model based on individual preference in China through machine learning algorithm. Sustainability (Switzerland), 13(14). https://doi.org/10.3390/su13147602
Rabani, M., Madessa, H. B., & Nord, N. (2021). Building retrofitting through coupling of building energy simulation-optimization tool with cfd and daylight programs. Energies, 14(8). https://doi.org/10.3390/en14082180
Moyano, D. B., Fernández, M. S. J., & Lezcano, R. A. G. (2020, May 1). Towards a sustainable indoor lighting design: Effects of artificial light on the emotional state of adolescents in the classroom. Sustainability (Switzerland). IPDM. https://doi.org/10.3390/su12104263
Response:
We are grateful to the reviewers for this comment.
The suggested references have been added. They are referenced at Lines26,27,51.
- Quantitatively justify whether the device has an ecological impact or energy savings.
Reply: Thank you for the comment.
Response:
We appreciate the reviewer for the suggestion.
Demonstrating the ecological impact and energy saving potential of the device is not the goal of current study. We will conduct further investigation in the future.
- Improve the quality of figures 3 ,4
Response:
We are very grateful to reviewers for their valuable suggestions on picture quality.
We have replaced Figure3 and Figure4.
Increase the size of figures 8,9,10,11 and 12
Response:
We are very grateful to reviewers for their valuable suggestions on picture quality.
We have adjusted Figures8, 9, 10, 11and12
Improve the quality of graphs 13,14,15,26 and 27; eliminating the background lines on the scale and in the frame.
Response:
We are very grateful to reviewers for their valuable suggestions on picture quality.
We have changed the images as required and replaced them.
Round 2
Reviewer 1 Report
The authors have significantly improved the manuscript. They introduced literature references to previously published figures and tables. The manuscript is suitable for publication.Author Response
We appreciate the reviewer for the recognition.
In the last revision of the manuscript, we changed some pictures to help readers understand our research work and express our research ideas more clearly.
Reviewer 2 Report
ACCEPT
Author Response
We appreciate the reviewer for the recognition.
Reviewer 3 Report
The authors have significantly improved the article.
Figures 26 and 27 should have the upper text "Simulated Result 1" and "Simulated Result 2" removed.
Author Response
We appreciate the reviewer for comment on pictures.
We have changed the images as required and replaced them.